# Midwinter melts in the Canadian prairies: energy balance and hydrological effects

Igor Pavlovskii[1], Masaki Hayashi[1], Daniel Itenfisu[2]

[1]Department of Geoscience, University of Calgary, Alberta, T2N 1N4, Canada
[2]Alberta Agriculture and Forestry, Edmonton, Alberta, T6H 5T6, Canada

*Correspondence to*: Igor Pavlovskii (ipavlovs@ucalgary.ca)

**Abstract.** Snowpack accumulation and depletion are important elements of the hydrological cycle in the Canadian prairies.
The surface runoff generated during snowmelt is transformed into streamflow or fills numerous depressions driving the focused recharge of groundwater in this dry setting. The snowpack in the prairies can undergo several cycles of accumulation and depletion in a winter. The timing of the melt affects the mechanisms of snowpack depletion and their hydrological implications. The effects of midwinter melts were investigated at four instrumented sites in the Canadian prairies. Unlike net radiation-driven snowmelt during spring melt, turbulent sensible heat fluxes were the dominant source of
energy inputs for midwinter melt occurring in the period with low solar radiation inputs. Midwinter melt events affect several aspects of hydrological cycle with lower runoff ratios than subsequent spring melt events and due to their role on the timing of the focussed recharge. Remote sensing data have shown that midwinter melt events regularly occur under the present climate throughout the Canadian prairies indicating applicability of the study findings throughout the region.

## 1 Introduction

The interplay between snow cover and frozen ground has a strong influence on runoff generation, infiltration, and associated processes such as groundwater recharge in cold regions (Lundberg et al., 2015; Tetzlaff et al., 2015). These processes are sensitive to changes in energy inputs, which govern snowmelt rates and, by extension, the partitioning of hydrologic outputs between streamflow and evaporative losses (Barnhart et al., 2016; Wang et al., 2016). A shift towards earlier snowmelt dates has been observed in mountainous areas throughout the world (Stewart, 2009), and even earlier melts are expected in
response to the climate warming (Rauscher et al., 2008). Net radiation is the primary driver of snowmelt even in periods with high sensible heat flux inputs (DeBeer and Pomeroy, 2017; Fayad et al., 2017; Hayashi et al., 2005) or during rain-on-snow events (Mazurkiewicz et al., 2008). However, these observations pertain to the melts occurring in March or later – a period with rapidly increasing solar radiation in the northern hemisphere. The roles played by different energy sources are not clear during midwinter melt events, which can cause snow-cover depletion when net radiation is much lower or even negative in
mid latitudes. Further uncertainty concerns the effects of low energy availability in midwinter and, hence, the effect of slow snowmelt rates on runoff generation in environments that are prone to midwinter melt events under the present and future climate.

This study investigates the energy balance of midwinter melt events and their effects on hydrological processes in the Canadian prairies, where complete snow-cover disappearance in midwinter is commonly observed (e.g., Akinremi et al., 1996; Maulé et al., 1994). Snowmelt is responsible for the majority of surface runoff in this environment (e.g. Coles et al., 2017), which contributes to streamflow (Shook et al., 2015), water inputs to wetlands (Johnson and Poiani, 2016), and groundwater recharge (van der Kamp and Hayashi, 1998). These conditions are similar to those of the Eurasian steppes, where snowmelt plays an important role in the hydrologic cycle (Barnett et al., 2005). Trends towards earlier snowmelt have been observed in both environments (Burn et al., 2008; Zhou et al., 2017), implying that observations made in the Canadian prairies will have a broader applicability in semi-arid cold regions around the world. Similarly to the mountainous areas, net radiation is a primary driver of spring snowmelt in the Canadian prairies (Granger and Male, 1978). In case of patchy snowpacks in addition to direct energy inputs, net radiation drives small-scale (intra-field) advection by warming snow-free areas above 0°C and, thus, prompting energy transfer towards remaining snow patches (Shook and Gray, 1997). Such energy transfer occurs as both sensible and latent heat fluxes (Harder et al., 2017).

The hydrological conditions in the Canadian prairies differ markedly from the mountainous areas despite the shared importance of snowmelt-driven hydrological processes. Relatively thin prairie snowpacks are more susceptible to depletion during midwinter melts compared to thick mountain snowpacks, and can undergo several cycles of accumulation and complete melt over one winter. The amount of water stored in the prairie snowpack is small relative to the soil moisture deficit, meaning that lateral water fluxes during snowmelt are driven not by filling of subsurface storage as in the mountains (Barnhart et al., 2016), but instead by the limited infiltrability of frozen soil (Granger et al., 1984) facilitated by both freezing and the low permeability of prairie soils derived from glacial tills and glaciolacustrine sediments covering most of the Canadian prairies (Fulton, 1995). As a result, the melt rate is hypothesised to directly affect the partitioning between runoff and infiltration with higher melt rates causing more runoff (Harder et al., 2018). In addition, numerous small depressions can trap snowmelt runoff in the prairies, and play an important role in the transformation of snowmelt runoff into streamflow by determining the hydrological connectivity in the landscape (Shook et al., 2015). These depressions are also important for groundwater recharge as they serve as conduits for the focussed recharge (van der Kamp and Hayashi, 1998). Thus, energy inputs during snowmelt and associated melt rate affect multiple elements in hydrological cycle in the Canadian Prairies.

The key objectives of this study are to: (1) determine the contributions of net radiation and turbulent heat fluxes to midwinter melts during the period with low solar radiation inputs, and (2) compare the effects of midwinter and spring melts on the hydrological processes driven by snowmelt runoff.

## 2 Study sites

Field studies were conducted at four instrumented sites located on the western edge of the Canadian prairies (Fig. 1, Table 1), in an area frequently affected by foehns in the vicinity of the Rocky Mountains, locally referred to as chinooks (Burrows, 1903; Nkemdirim, 1996), which were shown to drive snowpack ablation through sensible heat inputs (MacDonald et al.,

2018). The study sites have annual precipitation lower than potential evaporation (Sauchyn and Beaudoin, 1998) and extensive cover of glacial deposits (Fenton et al., 2013), which are common features of the Canadian prairies. The study sites have cold winters with monthly average temperatures below 0°C throughout November-March.

Despite general similarity, the climate varies among the study sites with annual precipitation ranging from 355 to 517 mm (Table 1). All four sites have the same seasonal pattern with most precipitation occurring in May-August. The temperature variation is less pronounced among the sites, with Calgary and Lethbridge being warmer than the other two long-term weather stations during winter months (Table 1).

The Spyhill, Stauffer, and Triple G sites have two closed topographic depressions regularly ponded by snowmelt runoff: C24 and GP at Spyhill; SE1 and SE2 at Stauffer; and W and E at Triple G (Fig. 2). Land covers were perennial grass for summer cattle grazing at the Stauffer, Triple G, and Fort Macleod; unmanaged perennial grass in the area around GP depression, and alfalfa in the area around C24 depression at the Spyhill site.

The present study focusses on the data collected in winters of 2015-2016 and 2016-2017, defined as the period with negative monthly average temperatures (i.e. November-March). Unless stated otherwise, the data were collected at the Spyhill, Stauffer and Triple G sites during both winters, and at the Fort Macleod site only in the winter of 2016-2017.

## 3 Methods

### 3.1 Meteorological measurements

Air temperature, humidity, radiation, and latent and sensible heat fluxes were measured at weather stations at all four study sites in half-hourly intervals (Fig. 2). Each site had an eddy-covariance system consisting of a krypton hygrometer (Campbell Scientific, KH20) and a sonic anemometer (Campbell Scientific, CSAT3) mounted at heights of 230-260 cm, and a temperature and relative humidity sensor (Vaisala, HMP45C) and a four-component radiometer (Kip & Zonnen, CNR1 or CNR4) at 195-210 cm. The eddy-covariance data were tilt-corrected using a planar fit algorithm (Wilczak et al., 2001), and further processed by applying the Webb-Pearman-Leuning (WPL) correction for vapour density (Webb et al., 1980) and for the separation between the krypton hydrometer and the sonic anemometer (Oncley et al., 2007). The energy balance correction was not applied due to the difficulty of quantifying the energy balance related to phase changes within the snowpack and underlying frozen soil. The measured vapor fluxes in g m$^{-2}$ s$^{-1}$ were converted to the reported latent energy fluxes in W m$^{-2}$ using the latent heat of sublimation at 0°C of 2834 J g$^{-1}$ (Hubbard and Neil, 2005).

The precipitation was measured with a weighing precipitation gauge (Geonor, T200B) equipped with an Alter shield at the Spyhill, Stauffer, and Fort Macleod sites; and at a weather station located 5 km north of the Triple G site. Precipitation gauges were equipped with anemometers to monitor wind speed required for the correction of wind-induced undercatch of precipitation. In the Canadian Prairies such undercatch in the absence of correction leads to an underestimation of annual snowfall by tens of percent (Pomeroy and Goodison, 1997). Therefore, all precipitation measurements at the study site were

corrected for undercatch following the procedure developed for a single-Alter-shielded precipitation gauges (Kochendorfer et al., 2017, Eq. 3).

## 3.2 Snow cover, surface ponding, and groundwater level monitoring

The snow water equivalent (SWE) of the snowpack was estimated by measuring snow depth at 1-m intervals along survey
lines (Fig. 2) using a metal ruler. Snow density was measured every 25 m at Fort Macleod and every 50 m at the other sites by weighing samples collected using an aluminium snow sampler with an internal diameter of 70 mm (Dixon and Boon, 2012). Density values were averaged for each line and multiplied by mean snow depth along the line to obtain an estimate of average SWE. The use of mean depth and density values is justified by the lack of correlation between measured depth and density during individual surveys and in the dataset as a whole.

At the Triple G site a snow drift formed in winter of 2015-2016 around the intersection of the two survey lines. While the snowdrift had a limited spatial extent, it gave a positive bias to the average snow depth along the survey lines. To remove the bias the southernmost 8 m along both lines were omitted from the analysis with remaining 92 m used to calculate average snow depth. This procedure was followed in winter of 2016-2017, although snow drift was much smaller and the average depth along the 92-m section of survey lines generally matched with that along the entire lines.

Time-lapse cameras (Wingscapes) were used to monitor snow-cover condition and surface ponding (see Fig. 2 for camera locations) by taking five photographs each day at 2-hour intervals starting at 9:00 (GMT -6). Snow gauges (Forestry-Suppliers) consisting of graduated plastic plates were vertically installed within the lines of sight of individual cameras.

The snow cover was manually classified in each photograph as being either "continuous" (no snow-free patches on the ground), "discontinuous" (isolated snow-free patches), "sporadic" (isolated patches of snow on the mostly snow-free
ground), or "no snow". Additionally, photographs with snow gauges were used to determine water levels in depressions by manually reading water levels during periods with open water or with snow-free ice. This dataset was augmented by pressure transducers (Solinst, Levelogger 3001) installed in the depressions following ponding events. Water levels were converted to runoff volumes using depth-volume-relationships derived from high-resolution topographical surveys of the depressions. The estimated runoff volume was used to calculate the ratio of snowmelt runoff to pre-melt snowpack SWE (i.e. runoff
ratio), which was approximated by the ratio of the amount of snowmelt runoff to the amount of total precipitation since the preceding complete snow-cover depletion (see Results section).

Piezometers were installed in 2014 to monitor hydraulic heads under depressions at the Spyhill (depression GP), Stauffer (depressions SE1 and SE2), and Triple G (depression W) sites in addition to existing piezometers in the C24 depression at the Spyhill site. The piezometers were screened at depths ranging from 1 to 12 m. The water level in the piezometers was
monitored using pressure transducers (Solinst, Levelogger3001; InSitu, miniTROLL) and by manual measurements.

### 3.3 Regional data sets

In addition to on-site field data, external data sets were used to cover a larger area and a longer period, and to provide the necessary context for local-scale, site-specific findings. The regional extent of snow cover was delineated using the Interactive Multisensor Snow and Ice Mapping System (IMS) satellite data set derived from a large number of satellite images, which assigns daily pixel values corresponding to sea, sea ice, snow-free land, and snow-covered land (National Ice Center, 2008a, 2008b). The subset of data used in the present study covered period from January 1999 to January 2017 inclusive. The pixel size was 24 km for 1999-2005 and 4 km for 2006-2017. The absence of snow in January for at least one day is used as a rough indicator of midwinter melt occurrence for the following reasons. The absence of snow cover can result only from a preceding melt event causing complete snow pack depletion, as there is no evidence that snowpack formation in the prairies can be delayed until January. The length of a snow-free period is controlled by the delay between melt and next snowfall and, thus, does not provide any information about melt intensity.

Long-term variation in meteorological conditions was examined using monthly homogenised and adjusted air temperature (Vincent et al., 2012) and precipitation (Mekis and Vincent, 2011) data from long-term weather stations (Fig. 1). Unadjusted daily values from the same weather stations (Government of Canada, 2017) were used to fill the gaps in the monthly dataset and to calculate temperature degree-day values.

## 4 Results

### 4.1 Meteorological conditions

Air temperature was closely correlated among the study sites, showing warm spells starting with a sharp rise in air temperature at all the sites in both winters (Fig. 3). The winter of 2015-2016 was warmer of the two with only short periods of daily mean air temperature below -5°C after the beginning of February (Fig. 3a), whereas temperature as low as -20 to -25°C was observed in February and March of 2017 (Fig. 3b).

Precipitation had stronger inter-site variability than temperature (Table 2). During the winter of 2015-2016 precipitation at the Spyhill and Fort Macleod sites was more than 20 mm (> 25%) below the long-term average mostly due to relatively dry February and March. Precipitation was higher during the second winter at all four sites.

### 4.2 Snowpack dynamics

Multiple snow cover depletion events occurred during both winters (Figs. 4c-4i, Table 3), and none coincided with rain-on-snow events, which are rare in the region. Snow cover at the Stauffer site was least affected by the warm spells, but there was a noticeable local variability within this site, as SE1 and WS areas had less snow than SE2 (Figs. 4e and 4f). Snow covers at Spyhill, Triple G, and Fort Macleod sites were more sensitive to warm spells with multiple instances of complete snow-cover depletion in both winters.

The IMS dataset generally captured the timing of snow-cover formation and its final disappearance in comparison to field data (Figs. 4c-4i), however, it missed some complete midwinter snow-cover depletion events observed at the study sites, for example, January 2017 event at the Spyhill and Triple G sites. However, the IMS dataset always indicated the presence of snow when the snow cover was present at the study sites.

The SWE in the winter of 2015-2016 peaked prior to the January 26-30 melt event with values ranging from 27 mm (Spyhill) to 40 mm (Triple G) (Fig. 4a). Extensive snow drift accumulation within depression affected results of snow survey at the Triple G site on January 20. After the melt event the snow cover at the Triple G and Spyhill sites disappeared completely except for short periods of snow cover in February (Figs. 4c and 4g), which were too short-lived to be captured by manual snow surveys. At the Stauffer site SWE remained stable in February until the snowmelt in March (Fig. 4a). It

should be noted that Fig. 4a only shows the SWE values from the snow surveys and does not show zero values corresponding to the periods of absent snow cover, as snow surveys were conducted only when there was snow.

In the winter of 2016-2017 there were two SWE peaks at the Spyhill, Triple G, and Fort Macleod sites. The peaks occurred before January 17 - February 4 and February 11-22 melt events with SWE values varying between 15 and 30 mm at different sites. At the Stauffer site the SWE increased over winter and peaked at 37 mm before the February 11-22 melt event (Fig.

4e).

At the Spyhill, Triple G, and Fort Macleod sites, the measured SWE on a given day was generally close to the antecedent precipitation defined as the total precipitation that occurred since the end of the last preceding snow-free period (Fig. 5). The measured SWE values at the Triple G site were higher than the antecedent precipitation (Fig. 5), which may have been influenced by the large snow drift formation. At the Stauffer site with a persistent snow cover, the SWE was generally lower

than the antecedent precipitation. Thus, antecedent precipitation is not representative of SWE at the Stauffer site, where snowpack survived the longest accumulating effects of partial melts and blowing snow sublimation. In contrast, frequent complete snowpack depletions at other sites remove effects of the preceding blowing snow events and making antecedent precipitation a reasonable approximation of SWE at the time of melt.

### 4.3 Surface energy inputs

Daily average values of sensible heat flux were mostly positive (i.e. towards the surface) until early March at all study sites during both winters (Fig. 6a, 6c). A strong temporal variability in sensible heat flux was observed with a number of distinct spikes separated by periods with low values around zero. The spikes appeared to occur simultaneously across the study sites but their magnitude differed. The daily average net radiation was mostly negative throughout December-February in both winters (data not shown), but became positive on several occasions following snow-cover depletion and associated albedo

reduction. Daily average values of latent heat flux oscillated around zero in December-February in both winters with magnitudes mostly below 10 W m$^{-2}$ (Fig. 6b, 6d). The flux became more negative (i.e. away from the surface) during snowmelt events, but the magnitude of latent heat losses was lower than sensible heat inputs. The same pattern was observed

during snow-cover depletion events in January and March 2017 (Fig. 6d). No correlation between magnitude of daily sensible heat flux and mean daily air temperature was observed in January during days with positive mean temperature in any of the sites (data not shown).

The relative importance of sensible heat and net radiation is illustrated by a comparison of two melt events in January and March 2017 (Fig. 7). During the January event most of the snow-cover depletion occurred overnight between January 16 and 17 when sensible heat inputs at the study sites surpassed 100 W m$^{-2}$ (sufficient for melting 1 mm of SWE per hour) (Fig. 7b). In contrast, net radiation before the melt was positive only for a few hours per day and peak values were below 25 W m$^{-2}$ (Fig. 7c). Following the melt, peak radiation values increased to > 100 W m$^{-2}$, but the positive radiation period was limited to a few hours per day (Fig. 7c). The second event caused rapid snow-cover depletion at the Spyhill and Fort Macleod sites on March 13 and slower depletion at the Stauffer and Triple G sites starting on March 14. The high sensible heat inputs were observed only at the Spyhill and Fort Macleod sites (Fig. 7e). In contrast to the January event, pre-melt net radiation reached as high as 80 W m$^{-2}$ and was positive for ca. 8 hours per day (Fig. 7f). Snow-cover depletion resulted in a rapid increase of the daily peak values of net radiation to above 300 W m$^{-2}$ at the Spyhill and Fort Macleod sites, followed by the Triple G site a day later, while it slowly increased over several days at the Stauffer site. The January event coincided with an increase in the magnitude of latent heat flux from slightly negative (up to -10 W m$^{-2}$) to peak half-hourly values of -35 W m$^{-2}$ at the Stauffer site, -70 W m$^{-2}$ at the Spyhill site, and -90 W m$^{-2}$ Triple G sites and -120 W m$^{-2}$ at the Fort Macleod site. Despite the high magnitude of negative latent fluxes during the melt event the net turbulent heat flux was positive (i.e. towards the surface) throughout the periods (data not shown). Similar peak values were observed at the study sites during the March event.

Relatively short snowmelt periods complicate the calculation of total energy inputs contributing to snowmelt due to high relative uncertainty in snowmelt timing. However, sum of positive energy fluxes over two day periods coinciding with disappearance of the majority of snowpack can be used to illustrate changes in the relative role of net radiation and sensible heat in driving snowmelt (Table 4). During midwinter melt the sensible heat inputs surpassed net radiation at all study sites by at least factor of 4. In contrast during spring melt net radiation surpassed sensible heat inputs by factor of 3 at the Spyhill and Triple G sites, factor of 2 at the Stauffer site, and was on par with sensible heat inputs at the Fort Macleod site. Note that Table 4 lists the sum of only positive energy fluxes (i.e. potentially contributing to melt), as simple sum of energy fluxes can be misleading (e.g. negative sum of net radiation over two day period does not mean that it does not contribute to melt during daytime). At the same, such notation masks effects of negative energy balance components on available energy and, by extension, on melt rates.

### 4.4 Depression ponding and groundwater response

#### 4.4.1 Stauffer

The ponding of the SE1 depression in the winter of 2015-2016 started on February 14 and ended on March 7 (Fig. 8b). Note that the soil under the depression had relatively high water content, close to saturation all year around. Several distinct runoff events were separated by periods of dropping water levels. The final water level recession started on March 8 at a rate of 14 mm $d^{-1}$. The hydraulic head beneath the depression started to rise on February 18 (i.e. four days after the ponding), reached a high level during the pond recession, and gradually fell after the disappearance of the pond (Fig. 8c). Time-lapse photographs indicated that the pond water surface was frozen on several occasions following the original ponding. However, liquid water was observed underneath the ice surface during water sampling events unrelated to this study.

The ponding of the SE1 depression in the winter of 2016-2017 occurred in two distinct stages with major runoff inputs on January 19 and February 14-16 (Fig. 8e), followed by a period with a stable water level until the recession started at a rate of 20 mm $d^{-1}$. The hydraulic head under the pond did not respond to ponding until March 24 (Fig. 8f) indicating the absence of infiltration. As in the preceding season, the hydraulic head reached a high level during the pond recession and gradually decreased after the pond disappearance. The ponded water in the depression completely froze after the original ponding during a cold spell in February (Fig. 8d). The ice surface was flooded by runoff during the February 14-16 melt event and froze during another cold spell in March with several centimetres of liquid water remaining between the two ice layers observed on March 14.

The pond in the SE2 depression was much smaller with a depth of a few centimetres in 2015-2016 (not shown in Fig. 8b) and ca. 10 cm in 2016-2017 (Fig. 8e). In 2015-2016 the ponding occurred on March 10 and only lasted for four days. In 2016-2017 two ponding events occurred on February 16 and March 19, but the formation of the surface ice layer prevented direct observation of the recession during the first event. An air-filled space was observed under the ice during a water sampling attempt on March 3 indicating that complete infiltration occurred by this time. During the second event the water level started dropping on March 26 at a rate as high as 50 mm d-1 (Fig. 8e). Pond disappearance in March 2016 and March 2017 coincided with a minor spike in hydraulic heads beneath the depression (Fig. 8c, 8f). In both cases a rise in the head was immediately followed by a recession. It was not possible to calculate runoff ratio at this site due to the high uncertainty in melt amounts during partial snow-cover depletion events.

#### 4.4.2 Triple G

Ponds formed in the depressions W and E in response to snowmelt runoff in both winters (Fig. 9b and 9e). In 2015-2016 major water level rises occurred on January 22, 27-28, and February 27-28 (Fig. 9b), coinciding with the depletion of snowpack (Fig. 4g). The water level declined at an approximate rate of 7 mm d-1 between the runoff events. The main water level recession started on March 6 and at a rate of 8.5 mm d-1. The hydraulic heads under depression W started to rise on

February 9 (18 days after the initial ponding), with highest values corresponding to the period of the main pond recession (Fig. 9c).

In 2016-2017 runoff inputs occurred on January 17, February 16, and March 18. (Fig. 9e), coinciding with the depletion of snowpack (Fig. 4h), and the water level did not decline between these events. The water level recession started on March 22 at an average rate of 14.5 mm d-1 until April 9 and 6.5 mm d-1 afterwards. The hydraulic head under the depression W started to rise on March 22 (64 days after initial ponding) and remained stable during the pond recession (Fig. 9f). As at the Stauffer site ponded water in the depression completely froze after the original ponding during a cold spell in February (Fig. 9d). The pond was flooded during the February runoff event and completely froze with no liquid water present in either depression on February 28.

The snowmelt runoff calculated from pond water volumes at the Triple G site did not correlate with the amount of antecedent precipitation (Fig. 10). In both seasons January-February melts tended to have lower runoff ratios (0.2-0.4) than March melts (0.6-0.8). The total runoff over the course of winter was 19.5 mm in 2015-2016 and 21 mm in 2016-2017. These values were calculated as the sum of ponded volume increases in depressions W and E divided by the combined area of their catchments.

### 4.4.3 Spyhill

No surface ponding was observed in GP and C24 depressions in the winter of 2015-2016, and no rise in hydraulic head occurred under these depressions (data not shown). Both GP and C24 depressions were ponded in the winter of 2016-2017 following the snowmelt event on March 15. However, in both cases the pond water level in the depression was less than 10 cm. The water depths in the depressions started to decrease around March 19 at a rate of ca. 10 mm d-1, which coincided with a rise in hydraulic heads beneath the depressions (data not shown).

### 4.5 Regional context

The IMS data covering the three Prairie Provinces showed the frequent occurrence of snow-cover depletion in January during 1999-2017 (Fig. 11). In some years (e.g. 2001, 2003, and 2006), the area of snow-cover depletion extended from the core of the prairies to the Foothills belt parallel to the Canadian Rockies, whereas in other years (e.g. 2002 and 2012) snow-cover depletion occurred mainly within the prairies core. The region around the Stauffer site underwent January snow-cover depletion only once (2006) during the 18-year period (Fig. 11). In contrast, snow-cover depletion occurred four times around the Triple G site, six times around the Spyhill, and all years except 2009 around the Fort Macleod site (Fig. 11). Additionally, all study sites had multiple days with snow cover in December throughout the 1999-2016 period supporting the use of snow-free period in January as an indicator of midwinter melts.

To examine regional-scale effects of meteorological factors, the occurrence or non-occurrence of January snow-cover depletion at the four study regions are plotted in relation to precipitation and air temperature measured at long-term weather

stations (Fig. 12). Total precipitation in December and January is used as an indicator of the potential amount of snow accumulation, and the sum of positive degree days (Fig. 12a) and the number of days with positive daily mean temperature (Fig. 12b) in January are used as an indicator of melt favourability. Whether midwinter melts are primarily driven by warm spell intensity (approximated by degree days) or by the occurrence of days with positive temperature alone is unclear due to strong correlation between these two metrics ($R^2$=0.78, where $R^2$ is the coefficient of determination). The distinction between the warm spell intensity (degree days) and positive temperature occurrence is important in the context of the lack of correlation in January data between magnitude of daily sensible heat flux and mean daily air temperature mentioned earlier. The lack of such correlation suggests that the increase in warm spell intensity does not promote midwinter melts by itself. Instead, apparent effect of warm spell intensity on midwinter melts (Fig. 12a) can be attributed to the association of this metric with positive temperature occurrence.

## 5 Discussion

### 5.1 Midwinter melts at the study sites and in the prairies

Snow-cover depletion events were observed at all study sites, but there were noticeable inter-site differences. The two sites located closest to the mountains (Spyhill and Fort Macleod) (Table 1) had more frequent and longer snow-free intervals than the other sites, highlighting the importance of foehns (chinooks) in causing snow-cover depletion. A similar decrease in foehn-driven melt intensity with the distance from the mountains was observed in the Alps (Zeeman et al., 2017). At a smaller scale, there was a noticeable variation in snow-cover depletion within the Stauffer site, where the snow cover depleted almost completely during warm events around the weather station (WS) exposed to wind from all sides while only minor reduction in snow cover occurred around the SE2 depression sheltered by a tree-covered hill on the west side (Fig. 4c, 4g). This is similar to the preservation of snow in topographically sheltered locations due to reduced advective heat inputs (Mott et al., 2013). However, while topographic sheltering affects relatively small snow patches, the effect of trees on wind speed can extend hundreds of metres downwind in case of a shelterbelt (Kort et al., 2011). Consequently, the intensity of midwinter melts is sensitive to conditions not only in the immediate vicinity but in a larger surrounding area as well.

The IMS data set missed the majority of partial and complete melt events observed at the study sites, similar to an earlier study (Brubaker et al., 2005) that reported the error of commission rate (failure to identify snow-free areas) surpassing the error of omission rate (failure to identify snow-covered areas) from December onwards. Nevertheless, despite the apparent undersampling of snow-cover depletion, the data set showed multiple such events not only in the "chinook belt" along the western edge of the Canadian prairies, where study sites are situated, but also in the core of the prairies, indicating that midwinter melts are common throughout the prairies under the present climate. Consequently, the mechanisms behind these events and associated effects on hydrological processes are important not only in the context of climate change, but for the interpretation of the past and present situations as well.

## 5.2 Energy inputs during spring and midwinter melt events

The data collected during snow-cover depletion events in January and March highlighted the difference in energy fluxes driving midwinter and spring snowmelt. Net radiation was the primary energy source for spring melt events, augmented by sensible heat inputs (Table 4). The daily peaks of net radiation were comparable with those of sensible heat flux even prior to the melt event, when surface albedo was still high (Fig. 7f). Furthermore, increase of net radiation after partial snow-cover depletion may have accelerated the melt through small-scale advection of sensible and latent heat from the low-albedo snow-free areas to the remaining snow patches (Harder et al., 2017; Shook and Gray, 1997). These observations confirm the importance of net radiation in controlling melt rates in open environments (Pomeroy et al., 1998). In contrast, January melt event was nearly solely driven by sensible heat inputs with most of the melt occurring overnight, when net radiation was negative (Fig. 7c). The observed change in the dominant energy source over the course of winter is consistent with previous study in mid-latitude open sites indicating that midwinter melts are primarily driven by sensible heat inputs with net radiation becoming the dominant energy source during snowmelts later in the season (Koivusalo and Kokkonen, 2002). The minor role of net radiation during midwinter melt events is likely universal throughout mid-latitudes, as low solar angles limit shortwave radiation inputs, which are further reduced by the high albedo of snow surfaces (Male and Granger, 1981). Even in the absence of snow cover net radiation is positive only for few hours per day (Fig. 7c), limiting the potential for radiation-driven small scale-advection from snow-free patches during midwinter melt events. Consequently, sensible heat inputs due to the large-scale advection of warm air masses is the primary source of energy for the midwinter melt events, except in situations when snow-cover area shrinks to only a small fraction of the landscape towards the end of the melt event.

The effect of patchy snowpack on energy fluxes is an important source of uncertainty in the measured data. The increase in measured net radiation over the melt period is caused by the reduced albedo of snow-free areas and does not represent changes in net radiation inputs into remaining snow patches. Instead a portion of extra radiative inputs is transferred towards snow patches via small-scale advection (Shook and Gray, 1997). As the top the boundary layer above snow patches is usually below the eddy-covariance sensor height (Granger et al., 2006) the energy fluxes associated with small-scale advection are excluded from measured sensible and latent heat fluxes (Harder et al., 2017). However, as such heat transfer to snow patches is ultimately driven by net radiation inputs into snow-free areas, the lack of direct measurements of small-scale advection does not fundamentally change the interpretation of the available data. Another source of uncertainty is the footprint of the eddy-covariance measurements, which routinely extends upwind by the distance of more than 100 times the sensor height and changes with time (Horst and Weil, 1994). As a result, measured fluxes reflect surface conditions (including snow cover fraction) for different areas at different times. However, the consistency between the datasets from the four weather stations indicates that temporal variation in eddy-covariance measurement footprint for each individual weather station did not have a major effect on conclusions about relative contribution of sensible heat fluxes to the midwinter and spring melts (Table 4).

The primary role of sensible heat flux in driving midwinter melt events has direct implications for the modelling of snowmelt process. Net radiation inputs are mainly controlled by the latitude, slope angle, and aspect (factors invariant over time), and snow-surface albedo, which changes in a predictable manner as snowpack ages. In contrast, sensible heat flux is influenced by atmospheric conditions and parameters that have high temporal and spatial variability such as air temperature

and wind speed. These factors can cause large variability in the timing and intensity of snow-cover depletion on a local scale, as observed within the Stauffer site (Fig. 4c, 4g).

## 5.3 Hydrological implications of midwinter melts

Midwinter melts affect runoff generation, which in turn affects other components of the hydrologic cycle. The increase in runoff ratios over the course of the winter season (Fig. 10) means that the occurrence of midwinter melts tends to reduce the

total amount of snowmelt runoff by shifting melt to a period with lower runoff ratios.

The likely cause for the variation in runoff ratios is the difference in energy inputs (and, hence, melt rates) between spring melt driven by net radiation and midwinter melts driven almost solely by the sensible heat flux associated with large-scale advection. Higher melt rates in spring are more likely to surpass frozen soil infiltrability, which varies in a range of $10^{-3}$ - $10^{2}$ mm $h^{-1}$ in the prairie soils in croplands (van der Kamp et al., 2003). Alternatively, the change in runoff ratios between

midwinter and spring melts (Fig. 10) may possibly be caused by an increase in soil water content during midwinter melts. Elevated water content leads to pore blockage by refreezing and, thus, "limited" infiltrability (Granger et al., 1984). In this scenario, a series of midwinter melts can lead to a progressive decrease in soil infiltrability and, thus, an increase in runoff ratio during the spring melt. However, realisation of such scenario requires refreezing to occur before meltwater can drain from the topmost few cm of soil. Thus, consistently positive temperatures over extended period after midwinter melt (Fig.

7a) are likely to reduce the effect of pore blockage on the following melt events.

Midwinter melts affect hydrological processes on a watershed scale by generating multiple meltwater pulses instead of a single spring freshet (Fig. 8, 9) and obscures the definition of common metrics used for hydrological trend analysis, such as spring freshet discharge and pre-melt SWE. For example, downward trends in spring runoff volumes and spring peak flows were observed in a number of prairie watersheds in recent decades (Burn et al., 2008). Midwinter melts can contribute to this

trend by reducing the SWE available to generate runoff later in spring. Consequently, the stream discharge during spring freshet would appear to be lower in a year with midwinter melts even for the same total runoff.

Midwinter melts also influence groundwater recharge processes. The ponding of topographical depressions by surface runoff during midwinter melts caused depression-focussed groundwater recharge under the ponds indicated by rises in the hydraulic head (Figs. 8 and 9). However, the timing of recharge differed between 2015-2016 and 2016-2017 despite nearly identical

original ponding dates and total runoff volumes. In 2015-2016 the recharge occurred shortly after the ponding, while in 2016-2017 it was delayed by as much as two months due to the formation of the ice layer preventing infiltration during

subsequent melt events. This observation highlights the lasting effect of the weather immediately after midwinter melt (and runoff) events on groundwater recharge.

The observed effect of melt timing on the hydrological processes can be compared to the sensitivity of the latter to precipitation characteristics. Several hydrological processes are more sensitive to the precipitation intensity and frequency than to its amount (Owor et al., 2009; Trenberth et al., 2003). Similarly, hydrological implications of midwinter melts can be linked to changes in the in intensity and frequency of meltwater release during snowmelt. Additionally, parallels can be drawn between the effects of midwinter melts and winter droughts (winter seasons with snowfall well below long-term average). Similarly to midwinter melts, winter droughts in the Canadian Prairies are associated with reduced spring SWE and snowmelt runoff, despite lower than average temperatures than in non-drought years (Fang and Pomeroy, 2008). This indicates that snowmelt runoff can be reduced by both warmer (due to midwinter melts) and colder winters (due to reduced snowfall).

The importance of midwinter melts will likely be amplified by the ongoing climate change. Even in the absence of midwinter melts, higher winter temperatures reduce runoff generation (Fang and Pomeroy, 2007). The current climate trends in the Canadian prairies suggest an increasing likelihood of midwinter melts as decrease in number of consecutive frost days outstrips decrease in total number of frost days (Vincent et al., 2018), making periods favourable to snowpack preservation increasingly intermittent. This trend is likely to continue due to projected increase in winter temperatures throughout Canadian prairies (Shepherd and McGinn, 2003). The change is likely to be most pronounced in the areas outside "Chinook belt" rarely affected by midwinter melts under the present climate. However, the positive correlation between winter temperatures and precipitation may mean that runoff can temporarily increase due to the impact of higher snowfalls outweighing the impact of higher temperatures (Fang and Pomeroy, 2007).

## 6 Conclusions

Net radiation and sensible heat flux have differing relative contributions to snowmelt depending on the timing of melt. Midwinter melts at the study sites were nearly solely driven by the sensible heat inputs associated with large-scale warm air advection during foehn events, while spring melts were dominated by net radiation inputs. The midwinter melts were characterised by lower runoff ratios than subsequent spring events. The likely cause is lower heat inputs during midwinter events leading to a slower melt, which allows more water to infiltrate. Reduced runoff during midwinter melts lowers both streamflows during freshets and, by limiting water inputs into depressions, depression-focussed groundwater recharge. Additionally, presence of midwinter melts leads to multiple occurrence of runoff events in a winter. Such multiple meltwater pulses may limit the applicability of commonly used metrics such as spring freshet discharge or spring pre-melt SWE in hydrological trend analysis.

Midwinter snow-cover depletion events occur regularly throughout the Canadian prairies, and their effects need to be considered in the hydrological understanding of winter processes in the region. Furthermore, the applicability of the findings regarding energy sources and hydrological effects of midwinter melts is not limited to the Canadian prairies. Similar effects of midwinter melts are expected in the open agricultural landscapes throughout mid latitudes with an exception of wetter climates with frequent rain-on-snow events.

Overall this study highlighted the importance of representing the effects of different energy sources and mechanisms in hydrological models of cold regions that experience midwinter melt events under the present and future climate. The magnitude and timing of midwinter melt events have noticeable effects on runoff generation and groundwater recharge, but the complex mechanisms causing these effects are not clearly understood. Further studies will be needed to understand the physical processes linking the surface and groundwater systems to midwinter melt events and how they may respond to the climate warming, which may increase the frequency and magnitude of melt events.

**Data availability**

Measured values presented in Figures 3, 4, 6, 7, and 9, as well as snow density data are provided in a table form in the supplementary material.

**Acknowledgement**

We thank Shelby Snow, Brandon Hill, Polina Abdrakhimova, Aaron Mohammed, and Dave Rae for field assistance; Larry Bentley and Edwin Cey for helpful discussion; and Alberta Innovates, Alberta Environment and Parks, Alberta Agriculture and Forestry, and Alberta Energy Regulator - Alberta Geological Survey for funding and in-kind support. IP received scholarships from the Canadian Geophysical Union and CREATE for Water Security Program funded by the Natural Sciences and Engineering Research Council. Constructive comments by the two anonymous referees improved the clarity of the paper.

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

**Tables**

**Table 1 Average precipitation (Mekis and Vincent, 2011) and air temperature (Vincent et al., 2012) at long-term climate stations close to the study sites, and the distance to the mountains from the study sites.**

| Study site | Long-term climate stn. | 1976-2005 average | | | Distance to the mountains, km |
|---|---|---|---|---|---|
| | | Annual precip., mm | Nov-Mar precip., mm | January air temp., °C | |
| Spyhill | Calgary | 476 | 91 | -7.7 | 60 |
| Stauffer | Olds | 517 | 96 | -9.5 | 230 |
| Triple G | Gleichen | 355 | 58 | -10.4 | 140 |
| Fort Macleod | Lethbridge | 421 | 98 | -7.0 | 80 |

**Table 2 Precipitation at the study sites and long-term averages for weather stations indicated in brackets (Mekis and Vincent, 2011).**

| Study site | November-March precipitation, mm | | |
|---|---|---|---|
| | 2015-2016 season | 2016-2017 season | 1976-2005 average |
| Spyhill | 67 | 93 | 91 (Calgary) |
| Stauffer | 81 | 109 | 96 (Olds) |
| Triple G | 58 | 81 | 58 (Gleichen) |
| Fort Macleod | 63 | 83 | 98 (Lethbridge) |

**Table 3 Midwinter snow-cover depletion events at the study sites identified on time-lapse photos.**

| Warm spell interval (dd.mm) | | Study site | | | |
|---|---|---|---|---|---|
| | | Spyhill | Stauffer | Triple G | Fort Macleod |
| Winter 2015-2016 | 01.12-09.12 | Complete depletion | Partial depletion | Complete depletion | No data |
| | 26.01-30.01 | Complete depletion | No depletion | Complete depletion | No data |
| | 05.02-19.02 | No snow present | Partial depletion | Complete depletion | No data |
| Winter 2016-2017 | 20.12-24.12 | Partial depletion | Partial depletion | Complete depletion | Partial depletion |
| | 17.01-04.02 | Complete depletion | Partial depletion | Complete depletion | Complete depletion |
| | 11.02-22.02 | Complete depletion | Partial depletion | Complete depletion | Complete depletion |

**Table 4 Sum of positive net radiation and sensible heat inputs during midwinter and spring melt events. *- periods start and end at noon on shown dates**

| Study site | Midwinter melt | | | | Spring melt | | | |
|---|---|---|---|---|---|---|---|---|
| | Period* | Net radiation, kJ m$^{-2}$ | Sensible heat, kJ m$^{-2}$ | Net radiation to sensible heat ratio | Period* | Net radiation, kJ m$^{-2}$ | Sensible heat, kJ m$^{-2}$ | Net radiation to sensible heat ratio |
| Spyhill | 16-18 Jan 2018 | 1687 | 7072 | 0.24 | 13-15 Mar 2018 | 10527 | 3226 | 3.3 |
| Stauffer | | 553 | 8318 | 0.07 | 17-19 Mar 2018 | 8450 | 3691 | 2.3 |
| Triple G | | 1529 | 11506 | 0.13 | 14-16 Mar 2018 | 7115 | 2405 | 3.0 |
| Fort Macleod | | 1228 | 16605 | 0.07 | 13-15 Mar 2018 | 8011 | 7788 | 1.0 |

**Figures**

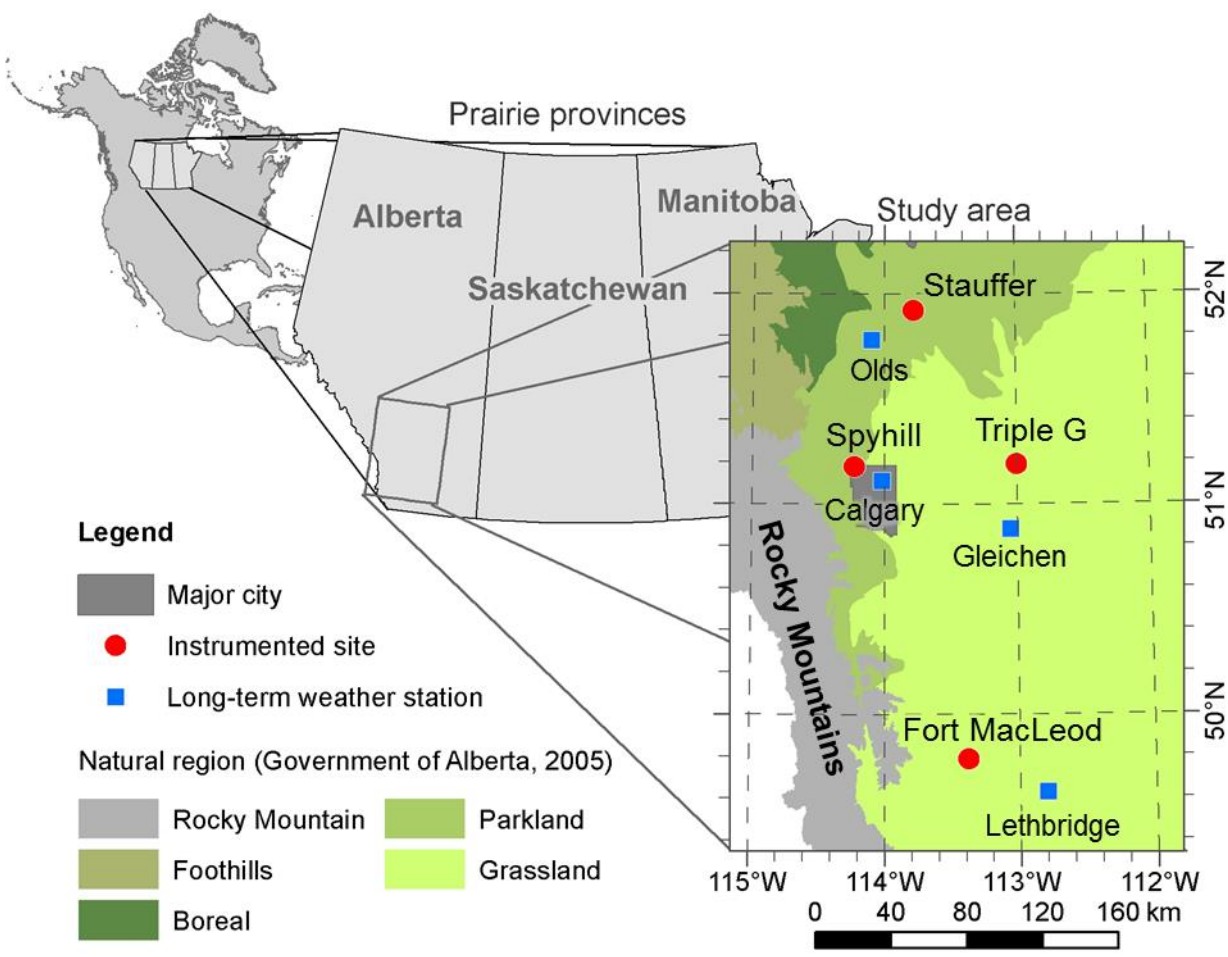

Figure 1: Location of the study sites. Parkland is a transition zone between semi-arid grassland and humid boreal natural regions.

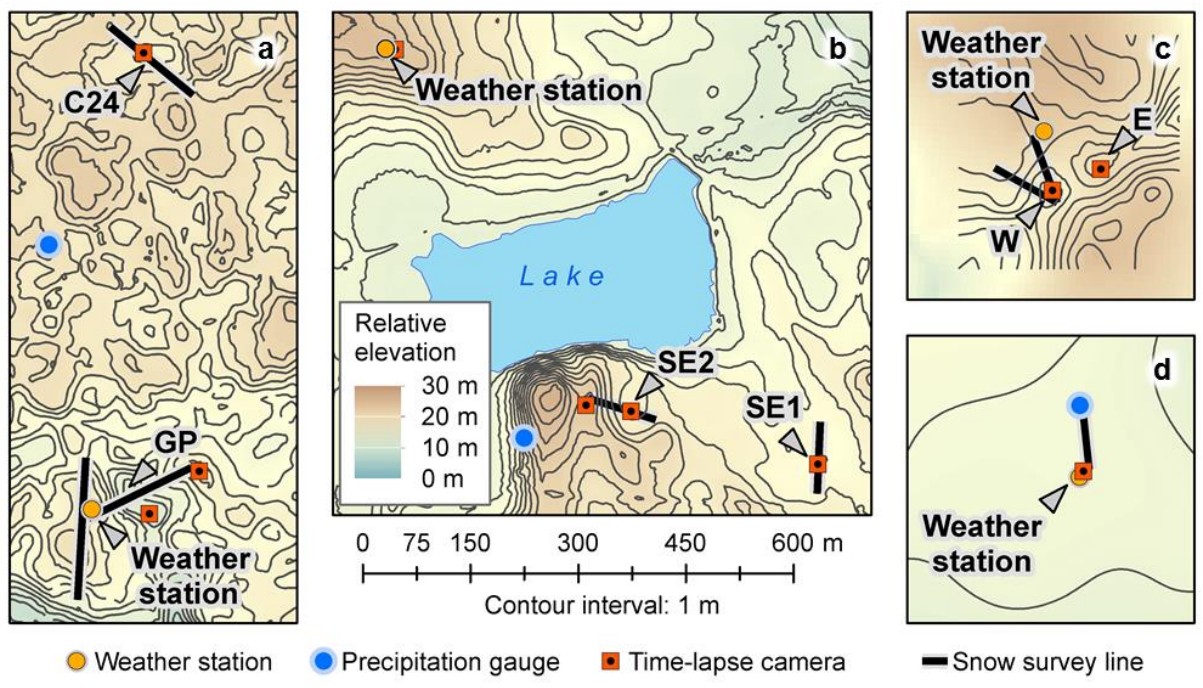

**Figure 2 Instrument set-up at the study sites: (a) Spyhill, (b) Stauffer, (c) Triple G, and (d) Fort Macleod. Contour interval 1 m.**

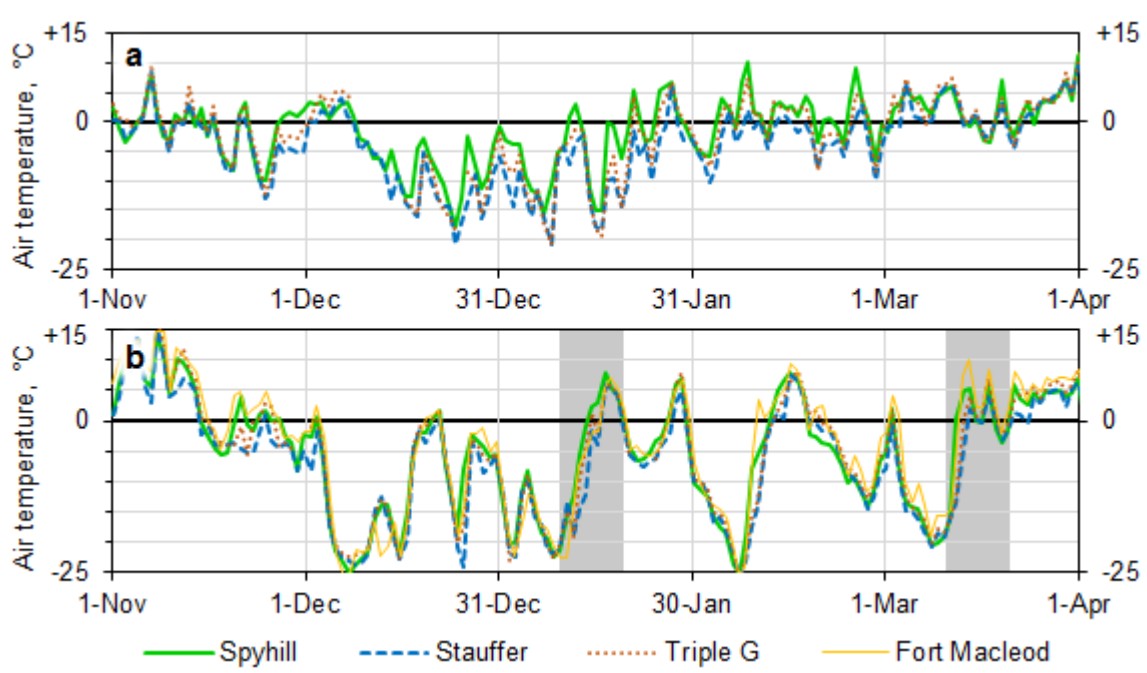

**Figure 3 Daily average air temperature at the study sites in the winters of (a) 2015-2016 and (b) 2016-2017. Shaded areas in (b) indicate events shown on Fig. 7**

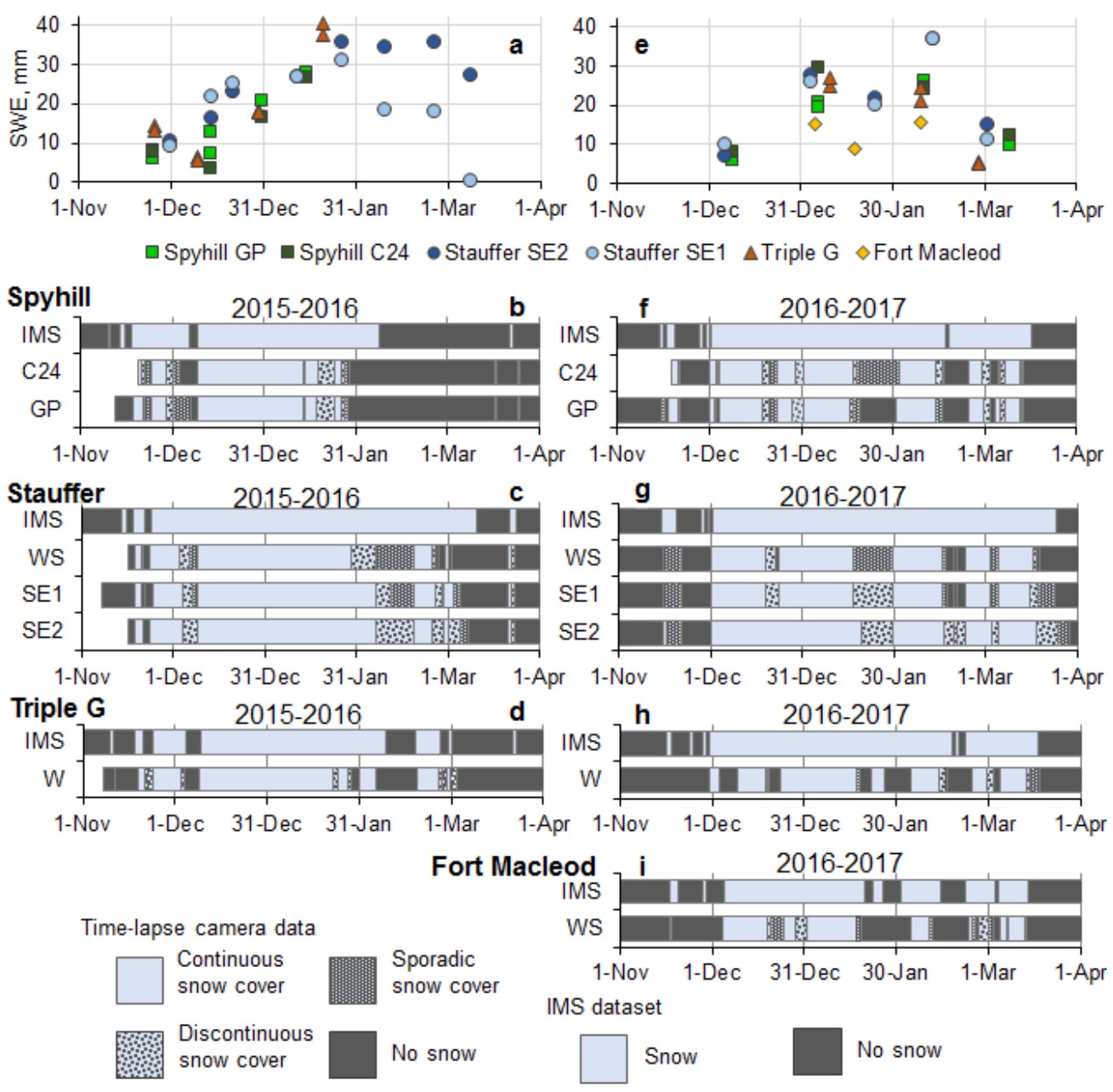

**Figure 4 Measured SWE and snow cover conditions at the study sites, derived from the IMS data set at 4 km resolution (National Ice Center, 2008a) and observed in time-lapse photos in winters of 2015-2016 (left) and 2016-2017 (right). Location of the sub-sites is shown on Fig. 3. WS – weather station.**

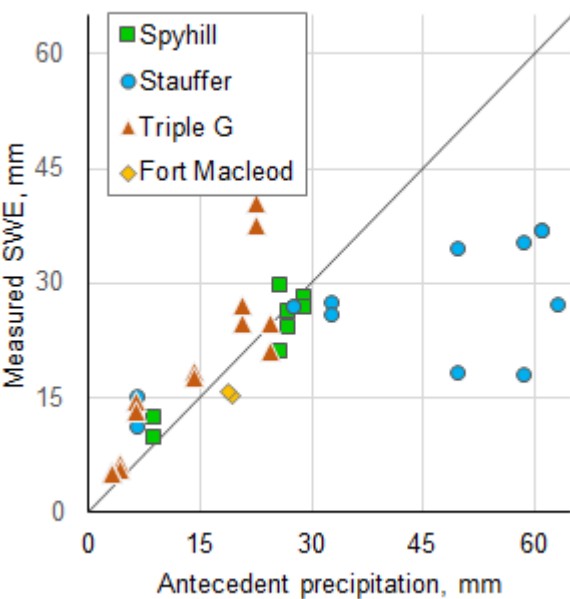

**Figure 5 Measured SWE at the study sites versus antecedent precipitation (total precipitation that occurred since the end of the last preceding snow-free period).**

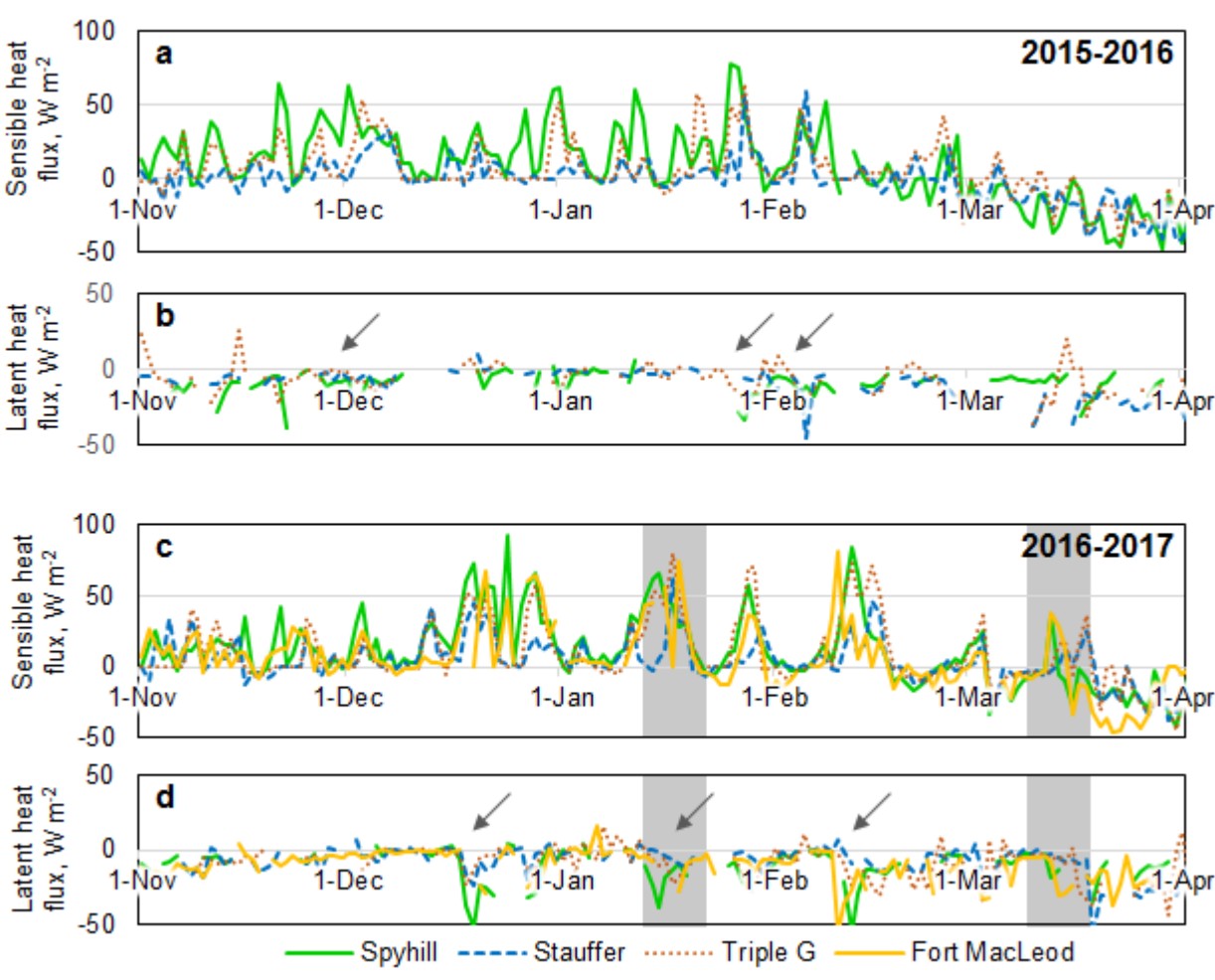

**Figure 6 Daily sensible and latent heat fluxes in winter 2015-2016 (a, b) and in winter 2016-2017 (c, d). Positive values indicate fluxes towards the surface. Arrows indicate midwinter snowmelt events at one or more study sites. Shaded areas in (c, d) indicate events shown on Fig. 7.**

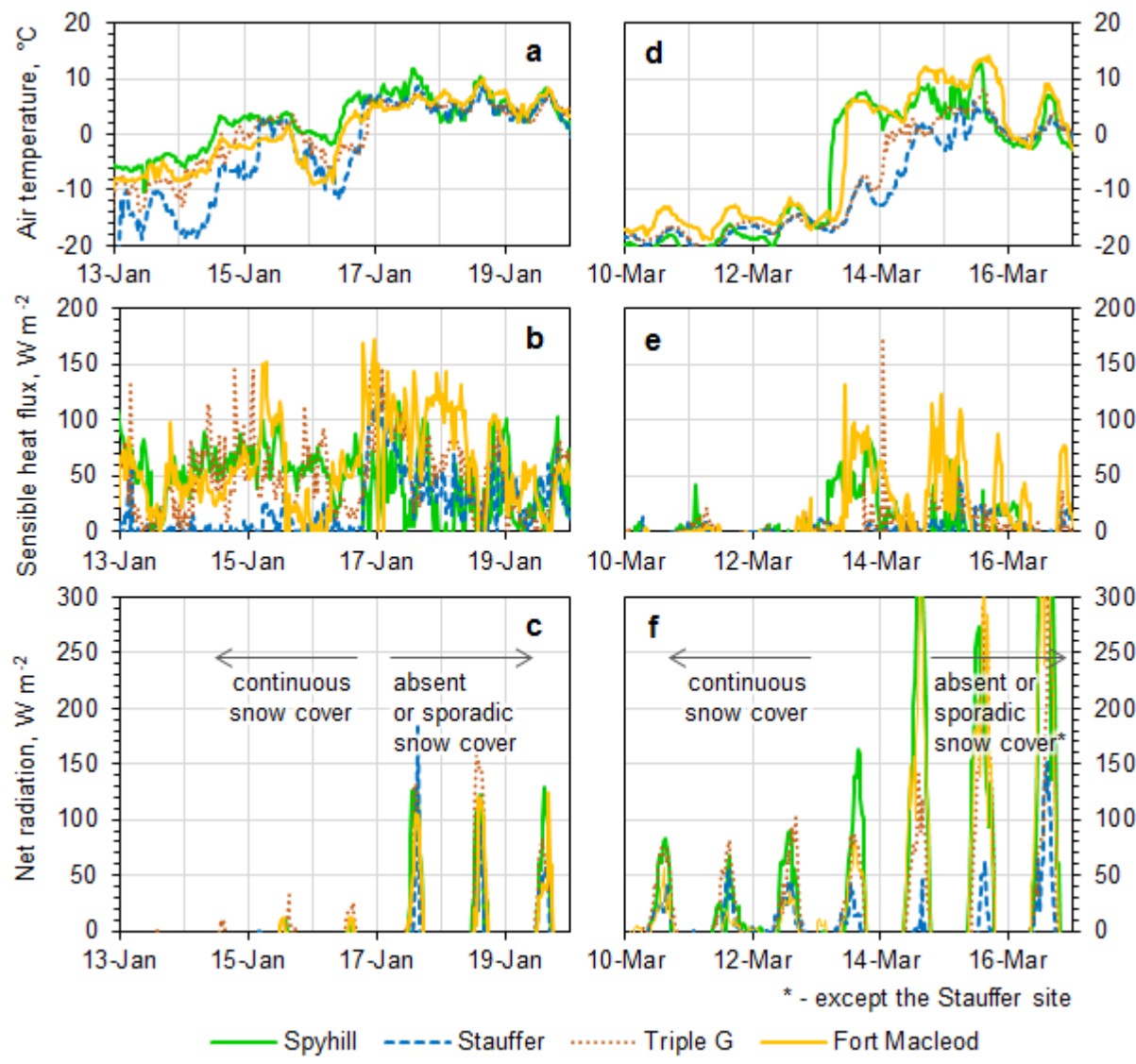

**Figure 7 Hourly temperature, sensible heat inputs, and net radiation during selected melt events in winter 2016-2017**

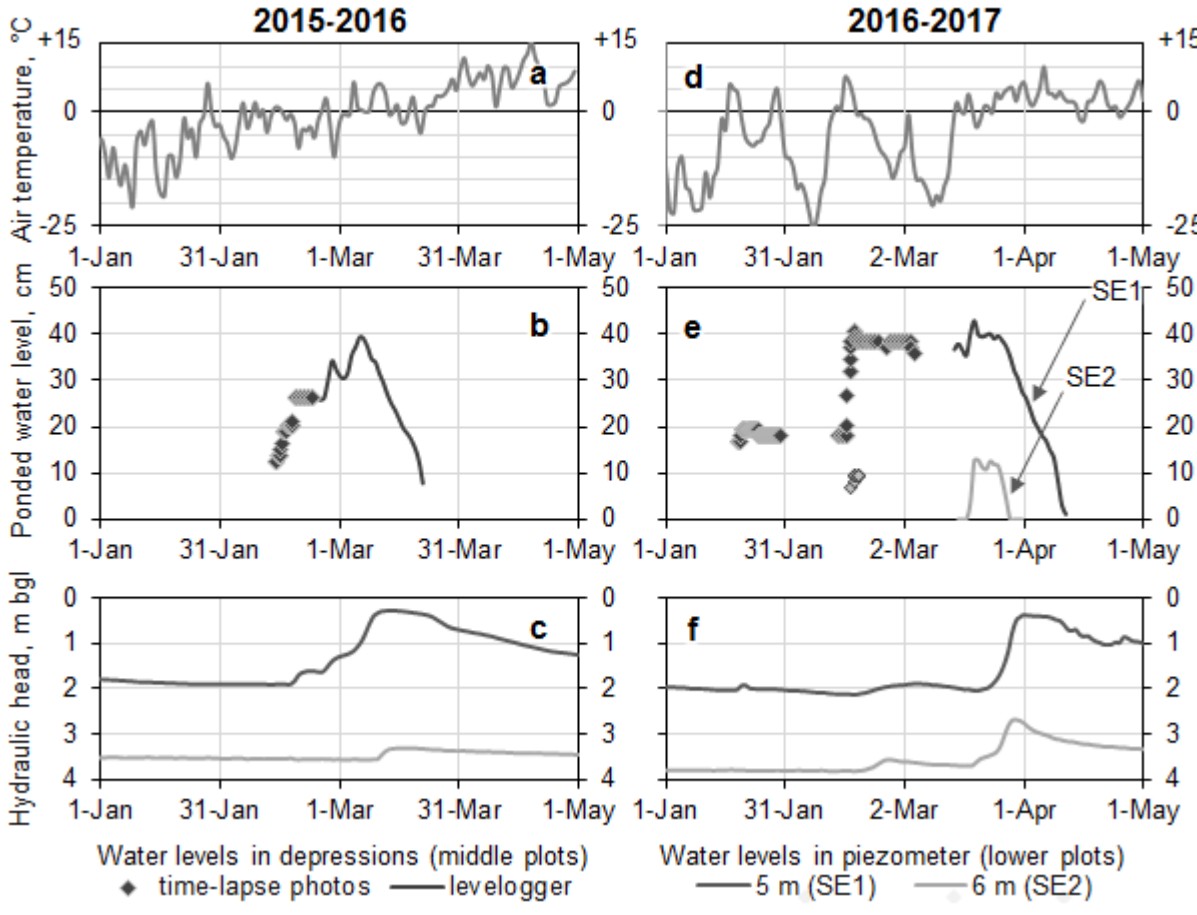

**Figure 8 Water levels in the depressions (above) and piezometers (below) at SE1 depression at the Stauffer site in winter 2015-2016 (left) and 2016-2017 (right). bgl – below ground level.**

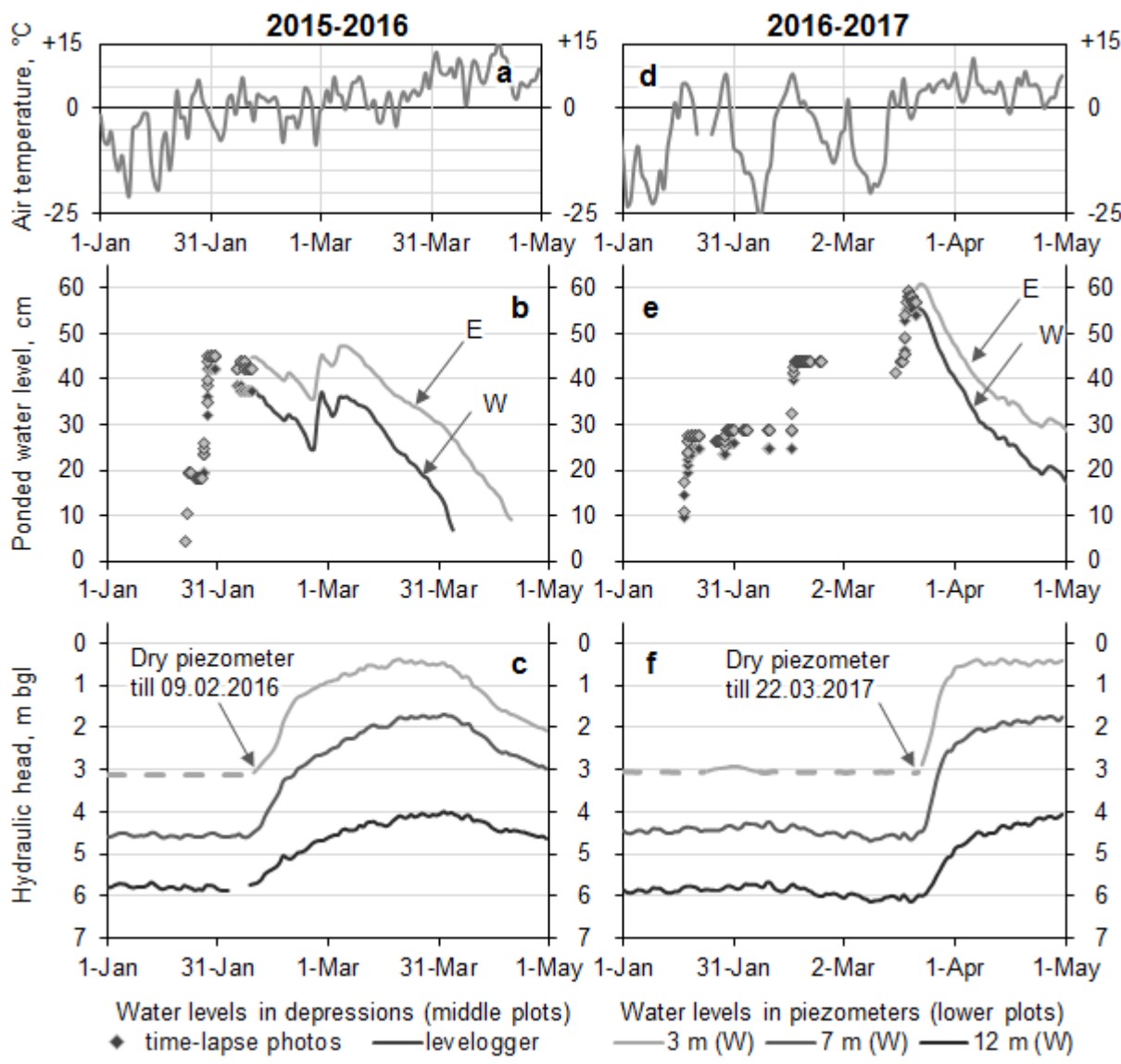

**Figure 9 Daily average water levels in the depressions (above) and piezometers (below) at the Triple G site in winter 2015-2016 (left) and 2016-2017 (right). bgl – below ground level. W and E refer to the two adjacent depressions as shown on Fig. 2.**

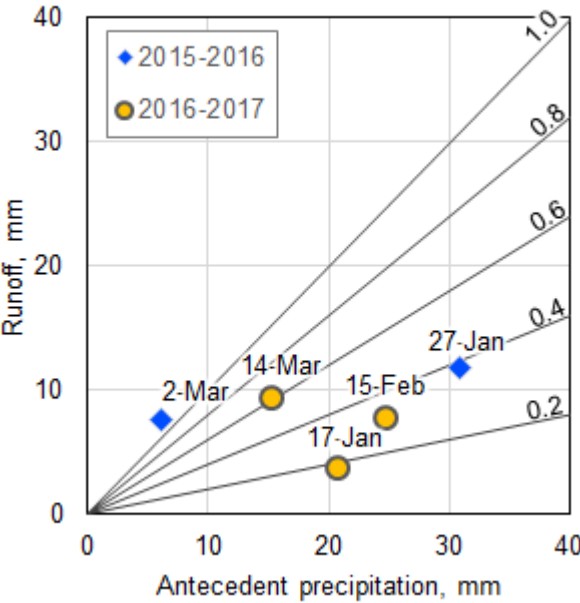

**Figure 10** Relationship between snow accumulation and runoff generation during snowmelt events at the Triple G site. Solid lines correspond to different runoff ratios. Labels above data points indicate the date of complete snowpack depletion. Antecedent precipitation is the total precipitation that has occurred since the end of the last preceding snow-free period.

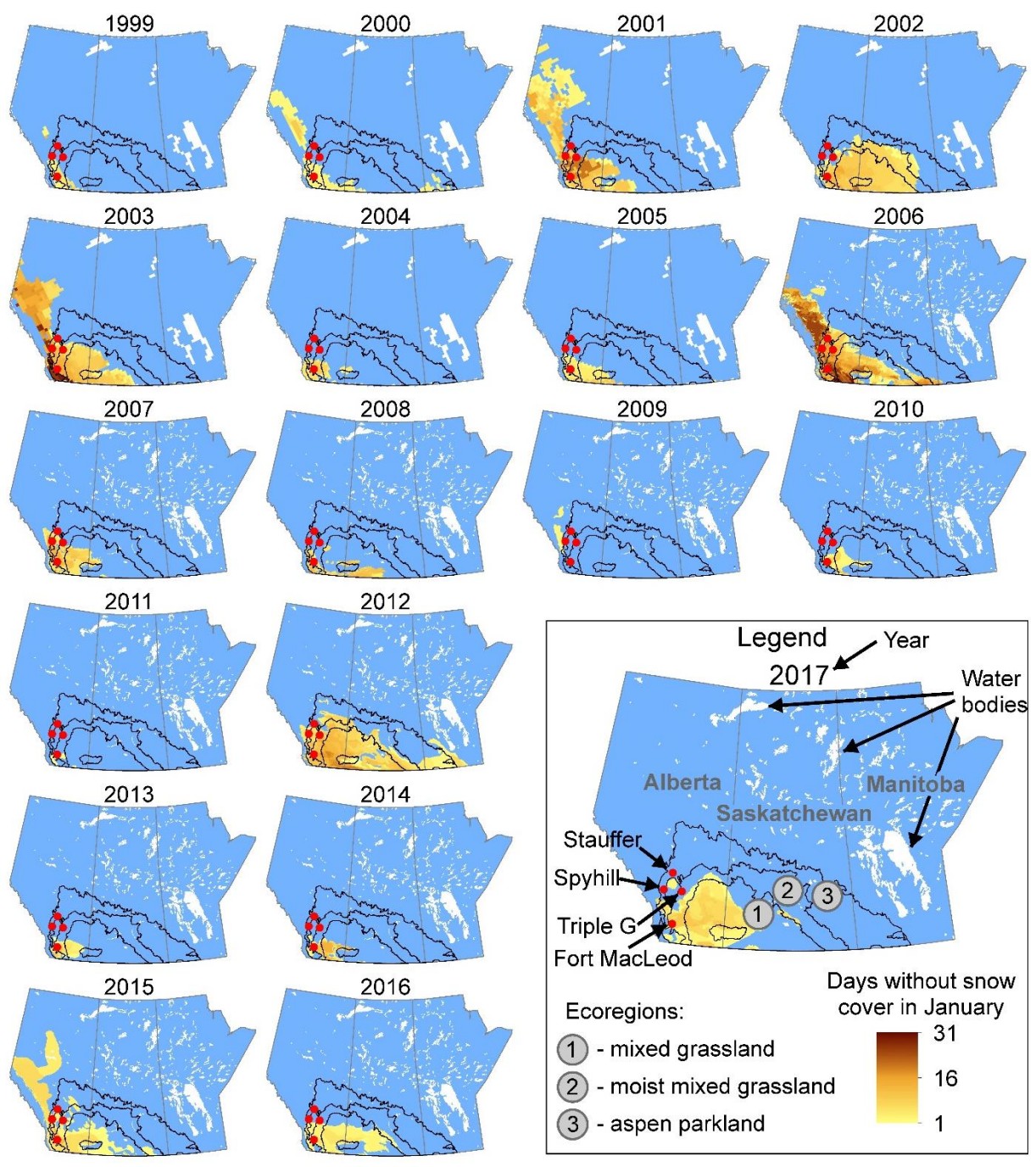

**Figure 11 The occurrence of January snow-free period in the Prairie Provinces derived from the IMS data at 24 km resolution (National Ice Center, 2008b) for 1999-2005 and at 4 km resolution for 2006-2017 (National Ice Center, 2008a). Ecoregion boundaries are from Agri-culture and Agri-Food Canada (2003).**

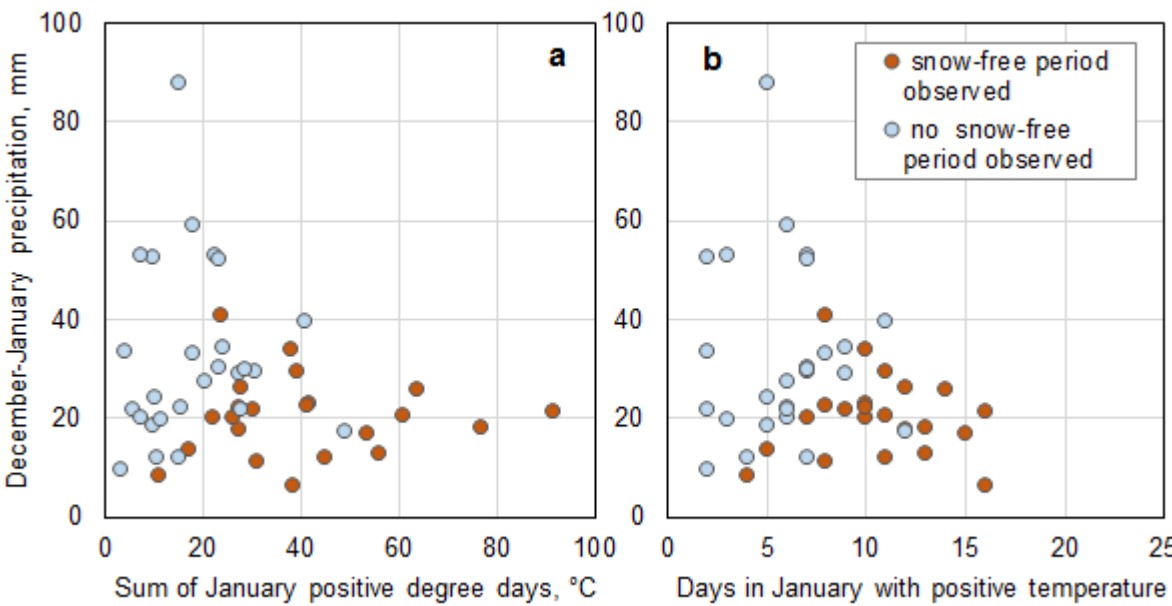

**Figure 12 Effect of January warm spells and December-January precipitation on the occurrence of midwinter melts at long-term weather stations (Calgary, Olds, Lethbridge, Gleichen).**