# Peer review of "Effects of midwinter snowmelt on runoff generation and groundwater recharge in the Canadian prairies"

_Hydrology and Earth System Sciences, 2018_

## Referee Comment (RC1) · Anonymous Referee #1 · 18 Sep 2018

This paper provides important insights into how infiltration occurs during winters in the Canadian Prairies. The results are likely to be of interest in cold regions throughout the world, in particular, semi-arid and arid ones. The paper is well-written and is in good shape in its current form. However, some minor revisions could help to improve its readability and increase its impact. My comments are as follows:

p.1, l.9: Be more specific about "the prairies" in this first line. Specifying Canadian would be helpful.

p.2, l.6: Semi-arid and arid is one component of this but a key factor in the Canadian Prairies is the presence of low permeability tills and clays. I think this might be

important when considering where else these results can be applied.

p.2, l.12-13: Again, geology is an important consideration when comparing mountainous environments versus the Canadian Prairies. Mountainous areas of Canada are dominated by thin soils and fractured bedrock in some areas and extensive alluvial fans in others. The Canadian Prairies have thick till and clay sequences.

p.7, l.15: Is the lack of hydraulic response due to a lack of infiltration or due to unsaturated conditions, which might lessen hydraulic diffusion?

p.9, l.7: Why is the data not shown? It would be useful to allow the reader to form their own judgements on the data.

p.11, l.1: There is no description of prairie soils in the manuscript and there should be. A paragraph on soils and geology somewhere in the paper would be quite helpful.

p.11, l.4-6: Was pore blockage actually observed or is this inferred?

p.11, l.23-27: The idea that midwinter melt leads to more infiltration and less runoff is a key finding of this study. The paper would be more impactful if this conclusion was placed in the context of other studies that have examined this issue. I am thinking specificially of a Owor et al. (2009, ERL) and earlier ideas on climate change leading to more runoff and drier soils presented by Trenberth et al (2003, BAMS).

p.12, l.7: Was the future climate for the prairies ever discussed? A brief discussion of this issue could help to frame the importance of this work.

---

## Referee Comment (RC2) · Anonymous Referee #2 · 6 Nov 2018

Review of Effects of midwinter snowmelt on runoff generation and groundwater recharge in the Canadian prairies by Igor Pavlovskii, Masaki Hayashi and Daniel Itenfisu

This paper explores the occurrence and drivers of mid-winter snow melt in southern Alberta, Canada and the implications that this has upon runoff generation groundwater recharge. It highlights the importance of this often-overlooked phenomena and how it can have temporally lagged consequences upon the main snowmelt event. Significant field data has been collected and described in detail which makes it a unique and valuable dataset to address the objectives. Despite this I find that the article will need

some major revisions before I recommend its acceptance. Major comments are first articulated, followed by minor/specific comments.

Major comments: Level of analysis:

There is a lot of information in this paper that is still fairly raw. Details and context are critical but overall if the level of presented could be at a more synthesis level that would clarify the arguments. For example, a main point of this paper was identifying the different energy sources over the melt season. This was done in a qualitative manner with respect to time series of hourly energy fluxes. It would be much more compelling (and the data seems available to do this) if the authors could calculate the energy balances of the melt periods to quantitatively say how much energy for melt came from turbulent versus radiation exchange. Discuss in terms of the total energy (J) for periods of interest rather than instantaneous maximum rates (W m-2)

Citations:

There is a lot of information already on prairie snowmelt processes that seems to not be fully referenced. It would strengthen this paper to put it into a firmer context of what has already been published. Significant snowmelt research on the prairies goes back to the 1970's at the Division for Hydrology at the U of S– some of these papers are cited but many that are relevant are not.

What do energy balance terms represent?

Turbulent fluxes measured with eddy covariance and radiation from net radiometers are used to quantify the energy balance terms. There is no discussion of what these observations represent as a snow cover depletes. The EC footprint will vary with wind speed and direction and will have variable contributions from both snow -covered and snow-free ground. Are these bulk averages therefore truly indicative of the energy contributions to snowmelt? As non-snow features increase contribution to the radiometer footprint a similar influence also occurs (but no subject to wind direction or speed like

the turbulent fluxes). These interpretation challenges for the energy balance observations need to be at least addressed.

Focus of paper?

The title of paper specifies that is "Effects of midwinter snowmelt on runoff generation and groundwater recharge in the Canadian prairies". Much of the paper considers the quantification of the energy sources driving mid-winter melts while runoff generation and groundwater recharge questions seem to be more ancillary. I feel that the title and paper content are somewhat disconnected.

Advection over patchy snow

Both large-scale and small-scale advection is mentioned in this paper. This is only introduced in the discussion section and the differences are not clearly defined. This should be incorporated into the introduction more formally. There has been a lot of recent work that has tried to disentangle this phenomena that could benefit this discussion.

Mott, R., M. Lehning, and Megan Daniels. 2015. "Atmospheric Flow Development and Associated Changes in Turbulent Sensible Heat Flux over a Patchy Mountain Snow Cover." Journal of Hydrometeorology 16: 1315–1340. doi:10.1175/JHM-D-14-0036.1.

Mott, R., S. Schlögl, L. Dirks, and M. Lehning. 2017. "Impact of Extreme Land Surface Heterogeneity on Micrometeorology over Spring Snow Cover." Journal of Hydrometeorology 18 (10): 2705–2722. doi:10.1175/JHM-D-17-0074.1.

Schlögl, Sebastian, Michael Lehning, and Rebecca Mott. "How are turbulent sensible heat fluxes and snow melt rates affected by a changing snow cover fraction?." Frontiers in Earth Science 6 (2018): 154.

Marsh, P., J.W. Pomeroy, and N Neumann. 1997. "Sensible Heat Flux and Local Advection over a Heterogeneous Landscape at an Arctic Tundra Site during Snowtnelt." Annals of Glaciology 25: 132–136.

Essery, R., R.J. Granger, and J.W. Pomeroy. 2006. "Boundary-Layer Growth and Advection of Heat over Snow and Soil Patches: Modelling and Parameterization." Hydrological Processes 20 (4): 953–967.

Liston, Glen E. 1995. "Local Advection of Momentum, Heat and Moisture during the Melt of Patchy Snow Covers." Journal of Applied Meteorology 34: 1705–1715.

Granger, R.J., J.W. Pomeroy, and J. Parviainen. 2002. "Boundary-Layer Integration Approach to Advection of Sensible Heat to a Patchy Snow Cover." Hydrological Processes 16 (18): 3559–3569.

Blowing snow

Blowing snow is a critical component of the winter time mass balance. This phenomenon needs to be addressed. Can it be discounted from time-lapse camera observations? If not the sublimation of blowing snow will challenge the comparison of runoff to antecedent precipitation.

Overall structure:

Paper needs more focus – less details. The science story is there: the turbulent flux energy source for mid-winter melts lead to more effective infiltration and the high radiation driven melt rates of the spring main event will have less effective infiltration and more runoff/depression focused recharge. The information presented should be limited to that which that support this main conclusion (if I interpreted correctly). In addition, the end of the discussion ends abruptly. If would be appropriate to add an additional section that discusses the implications of the mid-winter melt phenomena under climate change if there are any limitations to this type of speculation.

Specific comments

Abstract: ends abruptly with three separate conclusions. Is there a way to synthesize the conclusions better?

[Figure]

Page 1 Line 21-22: "hydrologic partitioning between streamflow and evaporative losses". What about infiltration or other sub-surface storage terms?

Page 1 Line 23: this is a prairie focused paper – is there a more appropriate citation for this?

Page 1 Line 29-30: unclear sentence

Page 2 Lines 1: snowmelt as main component of surface runoff is referenced in many other sources.

Page 2 Line 16-18: Many authors have looked at snowmelt energy dynamics/physics on the prairies – here is a small sample. While not specific to mid-winter melts these do provide a lot of physical understanding to describe snowmelt processes in the prairies.

MacDonald, Matthew K., J.W. Pomeroy, and Richard L.H. Essery. 2018. "Water and Energy Fluxes over Northern Prairies as Affected by Chinook Winds and Winter Precipitation." Agricultural and Forest Meteorology 248: 372–385. doi:10.1016/j.agrformet.2017.10.025.

Granger, R.J., and D.M. Gray. 1990. "Net Radiation Model for Calculating Daily Snowmelt in Open Environments." Nordic Hydrology 21: 217–234.

Norum, D. I., D.M. Gray, and D.H. Male. 1976. "Melt of Shallow Prairie Snowpacks: Basis For a Physical Model." Canadian Agricultural Engineering 18 (1): 2–6.

Harder, P., W.D. Helgason, and J.W. Pomeroy. 2018. "Modelling the Snow-Surface Energy Balance during Melt under Exposed Crop Stubble." Journal of Hydrometeorology 19 (7): 1191–1214. doi:10.1175/JHM-D-18-0039.1.

O'Neill, A.D.J, and D.M. Gray. 1973. "Spatial and Temporal Variations of the Albedo of Prairie Snowpack." Unesco-WMO-IASH Symposia 1: 176–186.

Gray, D.M., and P.G. Landine. 1988. "An Energy-Budget Snowmelt Model for the Canadian Prairies." Canadian Journal of Earth Sciences 25 (8): 1292–1303.

Granger, R. J., and D. H. Male. "Melting of a prairie snowpack." Journal of Applied Meteorology 17.12 (1978): 1833-1842.

Page 2 Line 19-21: Objectives do not reference runoff generation or groundwater recharge explicitly even though title does.

Page 3 Line 29-30: Any relationship between snow depth and density? See: Shook, K., and D.M. Gray. 1994. "Determining the Snow Water Equivalent of Shallow Prairie Snowcovers." In Eastern Snow Conference.

Page 4 Line 7: What are snow gauges?

Page 4 Line 15-17: How accurate how accurate is the delineation of the pond watersheds- assuming that this was used to multiply the precipitation to get total water volume.

Page 4 Line 28-30: Without a reference it is speculation to say that it is impossible for snowcover formation to be delayed to January.

Page 6 Line 9: Is this not indicative that there is blowing snow processes occurring (specifically sublimation) and this process should be addressed?

Page 6 Lines 11-21: How valid for snowmelt are your eddy covariance observations when SCA is < 1?

Page 7 Lines 2-5: Can one assume that high latent heat fluxes are related to snow surface exchange or could this also be from a separate source like ponded meltwater in non-snow areas?

Page 8 Lines 9-12: This sentence is confusing- please clarify.

Page 10 Lines 13-15: "studies" is plural but only cite one reference?

Page 10 Line 17-20: small scale advection may be limited by the duration of a positive net radiation but the advection flux can still be large, and depending on state of

snowpack and its SCA, and may lead to significant melt and SCA change. The authors should predicate this point with acknowledgment of this nuance.

Page 10 Line 28-31: difficult sentence

Page 11 Line 3-6: To put this into context of previous research perhaps stick with nomenclature of Granger et al. 1984 who termed this "limited" infiltration.

Page 11 Line 13: Wasn't this observed. Then why is this speculative with "may"?

Page 11-12 Conclusions: Portions seem to be rather repetitive from the end of the discussion?

Figure 3, 6 ,7: I find it difficult to get meaningful information with these hourly time series plotted for the four stations identified by colour. Hard to differentiate. Would it be possible to plot the total energy balance terms for the specific melt events? Overall figures have a lot of information and superimposed text which make them quite information dense.

Figure 11: this figure highlights that there are significant differences in mid-winter melts across the Canadian Prairies -likely due to the area of influence of chinooks. Perhaps this could be elaborated in the discussion to place these findings in the context of the broader Canadian Prairies and how the influence of mid-winter melts on runoff generation will vary across the region.

---

## Author Comment (AC1) · 2 Jan 2019

We thank the referee for constructive comments, which allowed us to improve the clarity of the manuscript. Please refer to the attached supplementary materials for the detailed response to the comments and revised manuscript.

Please also note the supplement to this comment:
https://www.hydrol-earth-syst-sci-discuss.net/hess-2018-423/hess-2018-423-AC1-supplement.zip

423, 2018.

---

## Author Response (AR1)

**Responses to editor's comments**

We thank the editor for his constructive suggestions. In the following editor's comments are typed in bold fonts, our responses in a regular font, and changes made in the texts in an italic font. The page and line numbers in our responses refer to those in the marked copy of the revised texts.

**Lack of considering this prior research also raises questions on the novelty of this research and the validity of its conclusions. For instance, you missed the comprehensive paper by MacDonald et al. (2008) http://dx.doi.org/10.1016/j.agrformet.2017.10.025 that described the turbulent fluxes and soil moisture changes in detail during chinook snowmelt events at several sites in Alberta.**

We agree that the MacDonald et al. (2018) paper is relevant to the present study. As such we have added reference to it (p.2 l.32). However, we have to note that midwinter foehn events described by MacDonald et al. (2018) do not involve neither snowpack disappearance, nor runoff generation.

**You also missed the paper by Fang and Pomeroy (Hydrological Processes DOI: 10.1002/hyp.7074, 2008) examining snowmelt runoff to prairie wetlands and the role of several hydrological processes in governing this runoff that are directly relevant to your study.**

We agree that the impact of midwinter melts has some similarity to one of the winter droughts. A small segment was added to discussion to convey this information to the reader (p.12 l.31-p.13 l.3):

*Additionally, parallels can be drawn between the effects of midwinter melts and winter droughts (winter seasons with snowfall well below long-term average). Similarly to midwinter melts, winter droughts in the Canadian Prairies are associated with reduced spring SWE and snowmelt runoff, despite lower than average temperatures than in non-drought years (Fang and Pomeroy, 2008). This indicates that snowmelt runoff can be reduced by both warmer (due to midwinter melts) and colder winters (due to reduced snowfall).*

**For instance, you did not correct for wind undercatch on your snowfall gauges and then claim that there was no blowing snow erosion at your site as you could close the mass balance. This is not credible as blowing snow events are well known in your research area Pomeroy and Li (JGR, 2000) as are substantial undercatch issues for snowfall gauges (Pomeroy and Goodison, {surface climates of Canada} 1997). By using corrected snowfall for wind undercatch you may find that your mass balance no longer can be closed and so you may wish to reconsider your conclusions regarding snow redistribution losses. Fang and Pomeroy (Hydrological Processes, DOI: 10.1002/hyp.73482009) discuss these in the context of snowmelt runoff to prairie depressions.**

Please note that the correction for the undercatch was performed for the data collected at the study sites (i.e. Figure 5 compares measured SWE with corrected precipitation values). However, we acknowledge that this wasn't stated clearly enough in the previous version of the manuscript. The corresponding segment was revised accordingly (new sentences underlined) (p.3 l.27 - p.4 l.2):

*The precipitation was measured with a weighing precipitation gauge (Geonor, T200B) equipped with an Alter shield at the Spyhill, Stauffer, and Fort Macleod sites; and at a weather station located 5 km north of the Triple G site. Precipitation gauges were equipped with anemometers to monitor wind speed required for the correction of wind-induced undercatch of precipitation. In the Canadian Prairies such undercatch in the absence of correction leads to an underestimation of annual snowfall by tens of percent (Pomeroy and Goodison, 1997). Therefore, all precipitation measurements at the study site were corrected for undercatch following the procedure developed for a single-Alter-shielded precipitation gauges (Kochendorfer et al., 2017, Eq. 3).*

**Importantly, I would like you to further discuss the role of midwinter melt events in creating ice layers in the soil (or at the base of the snowpack) that cause "restricted" infiltration - this would increase runoff, not decrease it and potentially invalidate your conclusions. Did you consider this in your calculations and did you observe formation of these ice or saturated frozen layers?**

The infiltration tests conducted at the study sites during relatively warm spell in late December 2016 indicated that soil at Triple G site after complete snow-pack depletion had sufficient infiltration capacity allow infiltration of >20 mm of water over time span of few minutes. I.e. warm temperatures after complete snowpack depletion allow the topmost few cm to drain before it infiltrated water refreezes, thus, preserving at least some infiltrability. While space constraints prevent us from adding relevant datasets to the present manuscript we acknowledge the need to inform the reader about alternative outcomes of warm spells. To address this we have added the following sentences to the discussion (new sentences underlined) (p.12 l.8-14):

*Alternatively, the change in runoff ratios between midwinter and spring melts (Fig. 10) may possibly be caused by an increase in soil water content during midwinter melts. Elevated water content leads to pore blockage by refreezing and, thus, "limited" infiltrability (Granger et al., 1984). In this scenario, a series of midwinter melts can lead to a progressive decrease in soil infiltrability and, thus, an increase in runoff ratio during the spring melt. However, realisation of such scenario requires refreezing to occur before meltwater can drain from the topmost few cm of soil. Thus, consistently positive temperatures over extended period after midwinter melt (Fig. 7a) are likely to reduce the effect of pore blockage on the following melt events.*

The infiltration tests results, as well as measurements by soil moisture and temperature sensors will be covered in the another manuscript (currently in preparation) which focusses on processes related to water flows through the frozen soil.

**In particular, you should relate the research findings to the findings of Fang and Pomeroy (2007) Snowmelt runoff sensitivity analysis to drought on the Canadian prairies, Hydrological Processes, DOI: 10.1002/hyp.6796 which specifically examined the impact of varying meteorological conditions on snowmelt runoff over frozen soils in a chinook-prone part of the Prairies and examined a climate change scenario for warmer and wetter winters.**
We agree.
We have added following sentences to the discussion (new sentences underlined) (p.13 l.4-12).

[revised manuscript text omitted]